# Functional Pyrazolo[1,5-*a*]pyrimidines: Current Approaches in Synthetic Transformations and Uses As an Antitumor Scaffold

**DOI:** 10.3390/molecules26092708

**Published:** 2021-05-05

**Authors:** Andres Arias-Gómez, Andrés Godoy, Jaime Portilla

**Affiliations:** Bioorganic Compounds Research Group, Department of Chemistry, Universidad de los Andes, Carrera 1 No. 18A-10, Bogotá 111711, Colombia; aj.arias@uniandes.edu.co (A.A.-G.); af.godoy@uniandes.edu.co (A.G.)

**Keywords:** antitumor scaffold, enzymatic inhibitory, *N*-heterocyclic compounds, organic synthesis, pyrazolo[1,5-*a*]pyrimidine, functionalization

## Abstract

Pyrazolo[1,5-*a*]pyrimidine (**PP**) derivatives are an enormous family of *N*-heterocyclic compounds that possess a high impact in medicinal chemistry and have attracted a great deal of attention in material science recently due to their significant photophysical properties. Consequently, various researchers have developed different synthesis pathways for the preparation and post-functionalization of this functional scaffold. These transformations improve the structural diversity and allow a synergic effect between new synthetic routes and the possible applications of these compounds. This contribution focuses on an overview of the current advances (2015–2021) in the synthesis and functionalization of diverse pyrazolo[1,5-*a*]pyrimidines. Moreover, the discussion highlights their anticancer potential and enzymatic inhibitory activity, which hopefully could lead to new rational and efficient designs of drugs bearing the pyrazolo[1,5-*a*]pyrimidine core.

## 1. Introduction

Pyrazolo [1,5-*a*]pyrimidine (**PP**) structural motif is a fused, rigid, and planar *N*-heterocyclic system that contains both pyrazole and pyrimidine rings [1]. This fused pyrazole is a privileged scaffold for combinatorial library design and drug discovery because its great synthetic versatility permits structural modifications throughout its periphery. The **PP** derivatives synthesis has been widely studied; thus, various reviews related to the obtention and later derivatization steps have been described in the literature [2,3,4,5], after the first critical review involving this attractive scaffold [6] (Figure 1).

Despite those reports, the synthetic transformations involving this motif still represent a research priority regarding the process efficiency, environmental impact, and the study of its multiple applications. These reports should address protocols that aim to minimize synthesis pathways, employ cheap reagents and develop processes that prevent or reduce waste production. Usually, **PP** derivative synthesis involves the pyrimidine ring construction via the interaction of *NH*-3-aminopyrazoles with different 1,3-biselectrophilic compounds such as β-dicarbonyls, β-enaminones, β-haloenones, β-ketonitriles, and so on (Figure 1b) [1,2,3,4,5,6]. Pyrazolo[1,5-*a*]pyrimidine scaffold is part of bioactive compounds with exceptional properties like selective protein inhibitor [7], anticancer [8], psychopharmacological [9], among others [10,11]. Furthermore, the biocompatibility and lower toxicity levels of **PP** derivatives have led them to reach commercial molecules, for instance, Indiplon, Lorediplon, Zaleplon, Dorsomorphin, Reversan, Pyrazophos, Presatovir, and Anagliptin (Figure 2) [1,2,3,4,5]. In recent years, this molecular motif has been a focus of study for promising new applications related to materials sciences [12,13,14,15,16,17,18,19,20,21], due to its exceptional photophysical properties as an emergent fluorophore [15,16,17,18,19,20,21]. Likewise, the tendency of pyrazolo[1,5-*a*]pyrimidine derivatives to form crystals with notable conformational and supramolecular phenomena [17,22,23] could amplify their applications towards the solid-state. Therefore, we aim to cover two main topics related to compounds bearing the pyrazolo[1,5-*a*]pyrimidine core. At the first one, the reader will find relevant synthesis strategies and functionalization reactions. Subsequently, in the second part, we focus on the recent compounds presenting antitumoral and enzymatic inhibitory activity. The examples described and commented herein came from the 2015 to 2021 period.

## 2. Synthesis and Functionalization

Firstly, we considered it convenient to carry out a schematic summary depicting the most relevant synthesis and functionalization sections (Table 1).

### 2.1. Synthesis

As stated before, the synthesis of **PP** derivatives is the focus of various research; however, more recent studies in this area are focused on improving known reaction protocols. Regardless, innovative synthesis methods still emerge that offer creative ways to modify established ones. Notably, the main synthesis route of pyrazolo[1,5-a]pyrimidines allows versatile structural modifications at positions 2, 3, 5, 6, and 7 via the cyclocondensation reaction of 1,3-biselectrophilic compounds with NH-3-aminopyrazoles as 1,3-bisnucleophilic systems (Figure 1b) [1,2,3,4,5,6,24,25].

#### 2.1.1. Biselectrophillic Systems

The synthesis starting from 1,3-dicarbonyl compounds is by far the most employed because it provides high functional group tolerance or enables conditions that are easily accessible while employing cheap commercial reagents. Acid and ethanol are commonly used solvents, while common catalysts include sodium ethoxide and amine-based bases [1,2,3,4,5,26,27,28,29]. This synthetic strategy is used to obtain large quantities of starting materials (ranging from mg to even kg) under simple conventional heating conditions.

In this context, Yamagami and co-workers developed a method to produce the 5,7-dichloro-2-hetarylpyrazolo[1,5-*a*]pyrimidine **3** via the cyclocondensation reaction of the 5-amino-3-hetarylpyrazole **1** with malonic acid (**2**) (Scheme 1) [30]. In this approach, the addition of POCl_3_ in the presence of a catalytic amount of pyridine produces an activated species of malonic acid phosphoric ester. This discovery led to the synthesis of **3** in a higher yield in a reduced time compared to the alternative conditions of dimethyl malonate under basic media as show by the authors. Moreover, this strategy eliminates the need for additional reactions that convert a hydroxyl group to chloride.

Similarly, the employment of β-diketones demonstrated to be effective for integrating fluorinated substituents at position 5, improving the electrophilic character of the 1,3-biselectrophile. Petricci and co-workers achieve the synthesis of **6**, where the carbonyl group substituent controls the reaction regioselectivity (Scheme 2a) [31]. Likewise, Lindsley and co-workers presented a protocol to obtain the 7-trifluoromethyl derivative **9** wherein the employment of heating by microwave (MW) irradiation significantly reduces the required reaction time. However, the brominated aminopyrazole **7a** is more reactive than **4** due to the electronic nature of the respective substituents (Scheme 2b) [32].

Considering aryl derivatives, Hylsová and co-workers employed arylketoesters **11a–b** to obtain the 5-arylpyrazolo[1,5-*a*]pyrimidines **12a–c** [33]. The authors do not report the complete protocol, though it is known that arylketones are a challenging substrate due to the lower electrophilicity of the carbonyl group (Scheme 3a). It would be interesting to evaluate conditions like toluene with tosylic acid [34]. Another report by Lindsley et al. described the synthesis of the 7-fluorophenyl derivative **14** using the brominated aminopyrazole **7b**. The acidic media and the electron-withdrawing effect from the halogen atom could promote the formation of products [35].

Furthermore, Fu and co-workers reported in 2020 a method to control the regioselectivity of the reaction. The reaction took place as expected with an excess of diethyl malonate (**15a**), and the pyrazolo[1,5-*a*]pyrimidine **16** is obtained in a 71% yield [36] (Scheme 4). However, adding **15** in stoichiometric quantity and heating the mixture neat produces the exocyclic amine from **10a** as the only available nucleophile. Hence, applying Vilsmeier-Haack (POCl_3_/*N*,*N*-dimethylaniline) conditions facilitates the annulation and subsequent hydroxyl/chloride exchange for delivering derivative **17**.

Conversely, if the 1,3-biselectrophilic system is a β-enaminone moiety, it enhances the reactivity or performance compared to 1,3-dicarbonyl compounds [1,37,38]. Portilla and co-workers employed diverse β-enaminone derivatives **19a–l** following a highly efficient methodology under MW irradiation, obtaining various 2,7-disubstituted products **20a–ac** in high yields (Scheme 5) [17,18,21,38]. The regioselectivity of the reaction can be controlled using the dimethylamino leaving group, where the initial condensation proceeds via an addition–elimination mechanism (aza-Michael type), thus bonding the NH_2_-group of the starting aminopyrazole with the Cβ of **19a–l** [37]. Successively, the cyclocondensation occurs by a nucleophilic attack of the pyrazolic nitrogen to the residual carbonyl group, where the subsequent loss of a water molecule leads to products **20a–ac** [1,37,38].

Following a similar approach, Guo and co-workers in 2019 reported the interesting structure series **23a–h** bearing various types of nitrogenous groups at position 7, which were obtained by using β-enaminone derivatives **22a****–h**. Notably, the authors achieved via compound **23d** the discovery and preclinical characterization of Zanubrutinib (BGB-3111): a novel, potent, and selective covalent inhibitor of Bruton’s tyrosine kinase. However, the obtained yields for intermediates **23a**–**h** were not found (Scheme 6) [39]. In other studies, the *N*-heterocyclic core has been synthetized from β-enaminone derivatives bearing aryl groups substituted with halogen atoms or methoxy groups. Additionally, carboxamides, aryl groups, nitriles, and esters have been employed as substituents on the starting *NH*-3-aminopyrazole [40,41,42].

Xu and co-workers reported an interesting method where 1-methyluracil (**25**), a heterocyclic β-enaminone, reacts with the aminopyrazole **24**. Through the uracil ring-opening induced by 24 and a later loss of a methylurea molecule, the 2-aryl-5-hydroxypyrazolo[1,5-*a*]pyrimidine **26** is produced. Employing an excess of **24**, it is possible to avoid the chromatographic separation for the purification of **26** (Scheme 7) [43]. Curiously, β-enaminones or, in general, the enone systems bearing a leaving group at β-position (e.g., Cl, OR, NR_2_, etc.) are synthetic analogous of 1,3-biselectrophilic ynones.

In a similar approach, Pankova and co-workers incorporated an alkyne group at position 7 of the fused ring (compounds **28a****–t**) by using the enone **27** having an ethoxy group at β-position [14]. Reaction yields are increased with electron withdrawer groups (EWGs) attached to the 4-aryl moiety in **27**, evidencing an electronic dependence on the nature of substituents. Likewise, the cyclization regiospecificity (enone vs. ynone moiety) emerges from the trimethylsilyl (TMS) group high electron-donating effect in π-electron systems (Scheme 8). This synthetic approach results in an attractive way to obtain the 7-alkynyl-2,6-diarylpyrazolo[1,5-*a*]pyrimidines **28a–t**, which later could undergo Pd-catalyzed carbon–carbon (C–C) cross-coupling reactions.

Moreover, the employment of enones for **PP** derivatives synthesis with two aryl substituents has been a matter of interest with different approaches. Adib and co-workers described using azidochalcones **30**, which yield milder conditions for synthesizing polysubstituted, products **31** in moderate to excellent yields (Scheme 9) [44]. The authors report short times for reactions and also established a recrystallization purification process. For these reactions, catalysis has been carrying out using ionic liquids [13] and nanoparticles [45], resulting in improved yields when the substituents were in the aryl group.

In the previous examples, enones bearing a leaving group at β-position were used as a 1,3-biselectrophile system. However, the reaction is often tricky when simple enones like chalcone derivatives are used, due to the unsaturation required in products [19,46]. Despite the inconvenience, this methodology provides valuable results for introducing aryl groups at positions 5 and 7 of the aza-heterocyclic core, with the advantage that starting materials are commercial or readily available by simple reactions (e.g., Claisen–Schmidt condensations). In this respect, Portilla and co-workers carried out the synthesis under rigorous (high temperature for a relatively long time) MW conditions of 2,5,7-tris(4-methoxyphenyl)pyrazolo[1,5-*a*]pyrimidine (**33**) starting from the substituted chalcone **32** and aminopyrazole **18d**. This synthesis allowed us to design and implement a probe for cyanide (CN^−^) sensing by a nucleophilic addition reaction on the carbon–carbon double bond of the receptor group and an intramolecular charge transfer (ICT) photophysical phenomenon (Scheme 10) [19].

Jismy and co-workers developed protocols for **PP** derivatives synthesis based on 1,3-biselectrophilic ynones [47,48,49]. Unlike previous approximations, employing ynones delivers to hydroxy/enone substituents at position 5 of the fused pyrazole. Thus, it is possible to obtain an addition acyclic intermediate **37** that, under basic media, delivers derivatives **36** (Scheme 11) [48]. The authors evaluate different conditions such as solvent, temperature, time, Lewis acid catalysis, and MW heating. This synthetic strategy produces compounds substituted at position 7 with inherently electrophilic groups such as CF_3_ (**36a–d**), which could be added in this manner, avoiding complex post-functionalization steps [49].

Comparatively, Schmitt and co-workers established a methodology that allows the functionalization with aryl or heteroaryl groups at position 7 (compounds **39a–f**) employing substituted alkynes **38a–f** (Scheme 12a). In addition, they took advantage of substituted pyrazoles to functionalize positions 2 and 3 (not shown) [50]. Recently, Akrami and co-workers developed a protocol capable of reduces reaction time employing dimethyl acetylenedicarboxylate (**38g**) and the aminopyrazole **21b** (Scheme 12b) [51]. The reaction was optimized in terms of solvent, temperature, and time, obtaining an ester group at position 7 of the respective product **39g**.

The use of other types of 1,3-biselectrophilic compounds for pyrazolo[1,5-*a*]pyrimidines synthesis have been described in addition to the works mentioned above. In this line, Hebishy and co-workers recently reported the synthesis of highly functionalized derivatives **42** [27]. By employing arylidenemalononitriles **41a–c**, it is possible to introduce an amine and a nitrile group in products **42a–c**, desirable groups for subsequent reactions such as carboxamides synthesis (Scheme 13). Similarly, Fouda and co-workers employ 2-(aryldiazenyl)malononitriles to obtain amines substituents at positions 7 and 5 [52].

Similarly, Portilla and co-workers developed an MW-assisted methodology to obtain 6-(aryldiazenyl)pyrazolo[1,5-*a*]pyrimidines **44a–q** by using 1,3-biselectrophilic derivatives bearing a hydrazone functional group at position 2, that is, reagent **43a–c** (Scheme 14) [53]. Remarkably, the synthesis depicted in Scheme 14 made it possible to introduce an amino group at position 5 of the heterocyclic core via a reductive azo bond cleavage [53]. Additionally, the electronic properties of the starting aminopyrazole **18** from various substituents at position 5 were evaluated. Notably, in this methodology, the authors report no solvent for the reaction, and purification requires little to no effort.

Furthermore, Zahedifar and co-workers reported the use of freshly prepared ketenes **45** in the preparation of compounds **46** from the corresponding substituted aminopyrazole **18a–e** under reflux in tetrahydrofuran (THF). The authors report high yields for the synthetized pyrazolo[1,5-*a*]pyrimidines **46a–e** and relatively short reaction times while that purification was carrying out through recrystallization (Scheme 15) [54].

#### 2.1.2. Multicomponent Reactions

Importantly, the pyrazolo[1,5-*a*]pyrimidines synthesis by multicomponent reactions have also been reported. In this context, the most common approximation resembles a Mannich reaction whose products usually undergo a later oxidation reaction. It is desirable to block any position which could lead to a side reaction, given that the derivatives obtained result highly substituted. Li et al. reported an approach using large quantities of the starting materials **47a** and **18** obtaining a dihydro derivative (not shown). Subsequently, the authors achieved an oxidation with DDQ without an intermediate purification step involved, to produce the unsaturated product **49** in 64% yield (Scheme 16a) [55].

Notably, the synthetic protocol of compound **49** is suitable for preparative quantities in conditions that are easily reproducible in the laboratory. Similarly, Shastri et al. developed a method focused on obtaining derivatives from different arylaldehydes **47a****–f**. In contrast to the previous example, the synthesis of products **51a****–f** contemplates mixing all the starting materials in the same vial (Scheme 16b) [56]. In this case, the more extended reaction periods could improve the yields, although aminopyrazole’s stereoelectronic properties could improve selectivity, avoiding side reactions like dimerization or aromatic substitutions. Besides, oxidation step relevance could be evaluated from work reported by Ismail, where a similar approach led to products in good yields, but the lack of an oxidation step produces **PP** dihydroderivatives (see Section 3.1 for detail) [57].

On the other hand, Jiang and co-workers reported an iodine-catalyzed pseudo-multicomponent reaction starting from aroylacetonitriles (β-ketonitriles) **52** and the sulphonyl hydrazine **53** [58]. The authors achieved derivatives **54**, and by involving two molecules of **52**, the positions 2 and 5 become substituted with the same aryl group. The reactions present moderate to good yields in a process that readily gives salts (Scheme 17). Notably, these compounds could have a longer shelf life compared with their analogs **54a–f**.

Likewise, Ellman et al. [59] designed a method based on catalysis with Rh complexes to obtain the pyrazolo[1,5-*a*]pyrimidines **58a–at** through the multicomponent reaction of aldehydes **55**, aminopyrazoles **56**, and sulfoxonium ylides **57** (Scheme 18). The synthesis of **58** was suitable for aldehydes with electron-donating groups (EDGs), heteroaryl, and haloaryl; however, enolizable aldehydes proved difficult substrates. The authors evaluated diversely substituted sulfoxonium ylides, obtaining good to excellent yields for **58ab–ah**. The reaction yield presents an increased susceptibility to pyrazole modification compared to the aldehydes and sulfoxonium ylides. Notably, the conditions optimized for the reaction gave a setup with benchtop materials, shorts reaction times, and high modulation options. Even though it is intended to be an MW heating protocol, conventional heating was also evaluated, providing good results. The scale-up of the reaction to 1 mmol yield 77% of the expected product using a lower catalyst charge (5 mol%).

Tiwari and co-workers prepared fused heterocycles **60a–c** through a C–C bond formation catalyzed with palladium [60]. The reaction proved efficient in generating the cyclic pyrimidine ring. However, only aryls without halogen or other exchangeable groups should be used by similar approximations reducing the scope of producing side products (Scheme 19).

#### 2.1.3. Synthesis by Pericyclic Reactions

Alternatively, pyrazolo[1,5-*a*]pyrimidines have been obtained by pericyclic reactions and without involving a starting aminopyrazole. In this respect, Ding and co-workers developed a protocol for the fused ring synthesis from acyclic precursors through a [4 + 2] cycloaddition reaction, which the authors report to be scalable and proceed in a one-pot manner [61]. The appropriate *N*-propargylic sulfonylhydrazone **61** is treated with a sulphonyl azide in the presence of catalytic copper (I) chloride since a click reaction drives the substrate **61** to a triazole formation, which discomposes to intermediate **62**. Subsequently, an intramolecular Diels–Alder reaction takes place, forming both rings of **63**. The dihydro derivative **63** could be treated in basic media to give the desired product **69** by an elimination reaction (Scheme 20).

#### 2.1.4. Synthesis with Fused Cores

Boruah and Nongthombam focused on developing fluorescent probes with biological activity by the androstenol derivatives **66** synthesis, which uses copper (I) iodide as a catalyst [62]. The synthesis proved to be efficient with yields ranging from 78 to 89%, where the pyrazolic nitrogen of **18** attacks the 1,3-biselectrophile **65** probably by a Michael type conjugated addition and the acetamide group as a leaving group (i). Subsequently, the cyclocondensation between the formyl (CHO) and amino (NH_2_) groups (ii) of the respective cyclization intermediate leads to the formation of **66a–e** (Scheme 21).

Comparatively, Mekky and co-workers carried out the synthesis of a bisbenzofuran derivative bearing two pyridopyrazolo[1,5-*a*]pyrimidine moieties (compound **68**). In addition to β-enaminone **19a**, the authors included β-dicarbonyl compounds or arylidenemalononitriles as 1,3-biselectrophilic systems, which were cyclocondensed with the fused *NH*-aminopyrazole **67**. The authors also evaluated the repercussion of MW heating achieving better yields and shorter reaction times (Scheme 22) [63].

Likewise, Jismy and co-workers evaluated a synthesis starting from the indazoles **69** and the appropriate alkyne **35d** to obtain the benzo-condensed derivative **70**, which possesses a CF_3_ and hydroxyl group on the pyrimidinic moiety (Scheme 23a) [64]. The authors also evaluated other positions for the bromo substituent in the amino-indazole, obtaining similar yields to that of **70** [64]. Similarly, Song et al. evaluated the obtention of the furan-fused product **72**, which is achieved by a Michael type conjugated addition over the enaminonitrile **71**, conserving the enantiomeric excess from the initial substrate (Scheme 23b) [65].

Ultimately, fused cycloalkanes to the **PP** core can be obtained, In this context, Elgemeie et al. developed various successful examples [27,66] by using trapped enolates (enone type compounds, **73** or **76**), which by a cyclocondensation reaction with *NH*-aminoprazoles (**40** or **75**), produces the tricyclic derivatives **74** or **77** (Scheme 24). The cycloalkanes to be fused vary from cyclopentane to cyclooctane, though these compounds are not functionalized in the provided examples. Additionally, the electronic effects on the starting pyrazole drive the reaction, showing, for example, better yields from **75** than those obtained from **40** against the same enone.

### 2.2. Functionalization

#### 2.2.1. Metal Catalyzed Reactions

##### Suzuki Couplings

This reaction is one of the most employed to add functionalized aryls on position 3 or 5; several organometallic species have been evaluated with favorable results, such as boronic esters, boronic acids, and fluoroboranes. The reactions have been carried out mainly using solvents like water mixed with some organic solvent (to maximize solubility) or in dioxane, and carbonates appear to be the preferred base, perhaps due to carbon dioxide formation facilitating the workup [67,68,69].

Jismy and co-workers designed a method for functionalizing **36e** at position 5 using a one-pot synthesis. They produce an exchange of hydroxyl group for a chloride atom by a NAS reaction; thus, employing optimized conditions for the coupling reactions could achieve aryls at position 5 like in product **79** (Scheme 25a) [48]. The reaction was further applied in the amine 3-bromosubstituted **81** and was optimized, finding the highly reproducible conditions to obtain **82** (Scheme 25b) [47]. The reaction shows an improvement in reaction times, and also, regarding the previous work, the authors found that modifying the ligand and catalyst enables the functionalization of the nucleophilic site at the pyrazolic moiety. Recently, they provided a method to add an aryl moiety at position 3 of the fused pyrazole **84**, which has a hydroxyl/enone group at position 5 and a CF_3_ group at position 7 (product **85**, Scheme 25c) [64]. The reaction conditions were screened to find an optimal setup, though the reaction conditions with the better performance are those previously reported. Interestingly, the fluorophenyl group and some heterocycles were found to be compatible with this strategy, with yields from 67 to 84% [64].

Related to this work, the employment of Suzuki coupling results is a common strategy in medicinal chemistry. Liu et al. reported the obtention of 17 examples about **88**, where the amines or ethers present at positions 5 and 7 are modified (Scheme 26a) [67]. Employment of PdCl_2_(dppf) as a catalyst and heating under MW irradiation appears to produce highly reproducible conditions for short-time reactions, and the authors obtained salt forms of each compound analogs to **88** for reactions with amounts above 200 mg of compounds related to **87** (Scheme 26a). Similarly, Lindsley and co-workers designed a fast reaction to functionalized the CF_3_–substituted PP **90** with aryls moieties bearing methoxy group (compound **91**) or fluor atoms (not shown) (Scheme 26b) [32]. Related to these advances, Drew et al. employed the same Pd-catalyst bearing dppf as a ligand to perform the coupling of **93** with the isoindolinone **92**, the reaction proceeds by the lability exchange with the halogen added according to the electronic properties of the position 5 against position 3 in the pyrazolo[1,5-*a*]pyrimidine **93** (Scheme 26c) [68].

The bromide atom left in **94** serves as a reactive center for a Buchwald–Hartwig coupling forming the acetamide moiety on **95** using BrettPhoss Pd G3 catalyst, a synthesis that is more efficient as stated by the authors. A related example to the metal-catalyzed C–N bond formation discusses in Section 4. The authors added the iodine atom at position 2, enabling the later addition of heterocyclic species at this position, delivering the 2,5-diheteroarylpyrazolo[1,5-*a*]pyrimidine **97** (Scheme 27a) [68].

This reaction is probably favored by the labile character on the I–C bond in **95** and the nucleophilic nature of its coupling partner **96**. Likewise, Harris et al. designed a method focused on the generation of disubstituted cyclopropanes, employing the proper borontrifluoride **98** and 3-bromopyrazolo[1,5-*a*]pyrimidine (**99**) which, under optimized reaction conditions, delivers **100** in 53% yield (Scheme 27b) [69]. Similarly, Lindsley and co-workers employed borontrifluoride derivatives as a coupling partner for Suzuki coupling reaction, adding a vinyl over position 3 of **136** (see Section Formylation Reactions for detail).

##### Sonogashira Couplings

This reaction commonly involves using a terminal alkyne bearing the **PP** core with an aryl/alkenyl halide as the coupling reagent. In almost all scenarios, Pd species are employed as the primary catalyst [38,70]. In this respect, Dong et al. designed a method to functionalize 5-ethynylpyrazolo[1,5-*a*]pyrimidine (**101**) with the alkyl bromide **102**. The reaction employs a copper salt with a ligand quinine derivative (L), achieving the formation of a new C–C (sp)/(sp^3^) bond in the absence of any Pd species [70]. The compound **103** is obtained in a high yield and with a high enantiomeric excess (*ee*), providing an approach towards functionalized products readily found in medically relevant molecules (Scheme 28).

Related to this matter, Childress et al. employed a common Pd catalyst and MW heating to achieve the functionalization of terminal alkyne **105** with an excess of 4-bromo-2-(trifluoromethyl)pyridine (**104**), obtaining the product **106** in 40% after purification by HPLC (Scheme 29a) [35]. The authors also obtain other two pyridine moieties in analogs of **106** employing the same methodology. Similarly, Jismy and co-workers developed a method to functionalized the aza-heterocyclic core at position 5 with wide scope [47,48,49,64]. Different from other authors, they generate the electrophilic coupling partner substituting the hydroxy/enone group at position 5 of **36d** (and analogs), then using the proper alkyne, the reaction delivers the product **108** (Scheme 29b) [49]. The reaction shows a high scope regarding the alkyne used, with a great influence on the substituent, wherein aromatic or conjugated ones achieve higher yields than alkyl or cycloalkyl substituents.

##### Other Metal-Catalyzed Reactions

Important reactions related to this matter are the C–H activations employing Pd species where the active species is prepared in situ. Bedford and co-workers developed a protocol that enables functionalizing of the positions 3 or 7 selectively in **PP**. An excess of aryl bromide (Ar–Br) was employed to obtain the 7-pyrazolo[1,5-*a*]pyrimidines **110a–f** a (Scheme 30) [71]. As expected, coupling reactions regioselectivity involving the heterocyclic core and an aryl bromide (Ar–X) depends on the electronic properties of reagents; indeed, the more π-deficient rings (e.g., Ar = pyrimidin-5-yl) behave well as electrophilic partners providing the 3-aryl derivatives **109a**–**f** in high yields via a coupling at the highly nucleophilic position 3 of **PP**. In contrast, π-excedent rings (e.g., Ar = 4-methoxyphenyl) behaves well when coupled to a more electrophilic place such as position 7, delivering **110a**–**f**. In order to explain these notable reactivity findings, the authors also report a DFT calculation analysis of the substrate, which are in agreement with those recently reported by Portilla et al. [18].

Similarly, Berteina-Raboin and co-workers designed a protocol to functionalize position 3 of the fused pyrazole **111**. The authors optimized the solvent, base, ligand, and palladium source; once the optimal conditions were founded, the authors proved various aryls bromines (Scheme 31) [72]. Furthermore, Gogula et al. provided a method to modulate the C–H activation of (sp^2^) or (sp^3^) carbon based on the temperature over the fused pyrazole **113** and analogs [73]. In addition, the added palladium generates a stable coordination complex **114** which the authors obtained, this species is responsible for the activation of the methyl C–H leading to a palladation (not shown) and subsequent formation of the C–C bond with the aryl iodine, delivering **115 a–f**. On the other hand, the activation of position 6 in the **PP** ring occurs by a tetramer compound (not shown), which generates a π-aryl palladation at position 6 and later arylation forming products **116a–f**. All reactions were carried out in hexafluoroisopropanol (HFIP) as a solvent, which proved to be efficient in terms of the compound’s achieved solubility and reaction temperatures. This solvent is known as a magical solvent for Pd-catalyzed C–H activation [74] (Scheme 32).

Finally, in the context of medicinal chemistry, McCoull and co-workers developed a method that enables the ring closure by olefin metathesis reactions on functionalized pyrazolo[1,5-*a*]pyrimdines **116** (Scheme 33) [75]. The protocol employs considerable catalysts quantities to achieve the reaction, and the authors evaluated various synthesis pathways to maximize the process efficiency. The stereochemistry over **116** correctly in the pyrrolidine fragment controls the final conformation achieve in **117**, which is an atropisomer due to restricted rotation by the aryl fragments of the macrocycle.

#### 2.2.2. Nucleophilic Aromatic Substitution Reactions

The nucleophilic aromatic substitution (NAS) reaction is common to functionalize with nucleophiles the positions 5 and 7 of the fused ring. The reaction results are widely employed in medicinal chemistry because it allows modifications with various structural motifs at electrophilic positions of the pyrimidine ring. Recently, it has been employed with the aim of adding aromatic amines [34,67], alkylamines [48,76], cycloalkylamines [55,77,78], and substituted alkoxides [67]. In this respect, McNally, Paton, and co-workers designed an interesting way of coupling the 5-(diphenylphosphanyl)pyridine **118** to the position 7 of the fused pyrazole **PP**, generating the phosphonium salt **119** according to the authors [79]. This approach opens the door to a new functionalizations with weaker nucleophiles. The authors report the importance of a strong acid in the medium for the reaction mechanism based on pyridines; however, the electronic properties of substituted pyrazolo[1,5-*a*]pyrimidines could enable the avoidance of this requirement. The treatment of **130** with a source of chloride provided **120**, although in a low yield (Scheme 34).

Similarly, Jismy et al. designed a methodology for the functionalization of position 5 employing PyBroP, probably with the formation of the intermediate species **121**, which facilitates the secondary amine **89** formation by the nucleophilic substitution of **122** over **121** (Scheme 35a) [47,49]. The authors employ this methodology in a one-pot manner achieving high yields. Additionally, the authors evaluated the efficiency of first obtaining **81** and then performing a Suzuki coupling reaction at position 3, obtaining **82** (see Scheme 25b above). As a result, the procedure done in that order delivers the final product in higher yields compared to the Suzuki coupling and then the aromatic nucleophilic substitution reaction [47]. Similarly, Berteina-Raboin developed a multicomponent method to obtain **112a** analogs, controlling the equivalents of added amine (morpholine, Scheme 35b); they were able to decrease the poisoning of the catalyst achieving good yields for various 7-aminoderivatives [72].

Additionally, Jiang and co-workers reported an efficient synthesis of various 7-(*N*-arylamino)pyrazolo[1,5-*a*]pyrimidines **125a–e** due to the electronic properties of the employed aromatic amines and the authors use two synthesis pathways (Scheme 36) [36]. Engaging triethylamine with amines coupled to electron donor groups are conditions that allow to deliver **125a–b** in high yields. In contrast, amines bearing EWGs require strong basic conditions, as is exemplified by products **125c–e**.

#### 2.2.3. Other Functionalization Reactions

##### Carboxamide Synthesis

Besides the use of amines to expand the structural diversity, a practical secondary way to expand moieties installed over the ring is the formation of amides. Zou et al. developed an important way to functionalize adjacent aryls in **126**, where the added catalyst participates with a C–H activation directed by the pyridine-like nitrogen on the pyrazole side of **126**. The electronic properties of **127** facilitate the addition of the amide fragment by the C–N bond formation with decarboxylation of **127** (Scheme 37). The authors tested the reaction on various substrates where, interestingly, both steric and electronic factors modulate the reaction. However, the halide derivatives could be challenging due to side reactions and electronic properties of the 7-aryl group. From the author’s perspective, the bond between the **PP** and the aryl group allows the amide formation and could be a promissory route to generate challenging amides.

A common strategy in the synthesis of amides over the **PP** core consists of using benzotriazole derivatives (HATU) to activate carboxylic acids. Manetti and co-workers reported use of ester **129** which, under basic conditions at room temperature, delivers the corresponding acid (not shown). Subsequently, with an appropriate amine, the weak nucleophilic carboxyl attacks the HATU and generates a species susceptible to the amine **130** at room temperature, obtaining **131**. The authors report the employment of this strategy to achieve diverse amide derivatives (Scheme 38a) [31].

Moreover, Lim and co-workers employed HATU, achieving the amide formation between the carboxylic acid **132** and the aminopyrazole **133**. For this reaction, it was necessary to block the pyrrolic nitrogen position on **133** due to its nucleophilic character (Scheme 38b). Other strategies used by the authors to develop amides on position 3 included the formation of acyl chlorides from carboxylic acids [80]. Related to this work, other authors have also employed a similar protocol to access amides [29,39,75,78,81,82].

##### Formylation Reactions

The procedure focused on the obtention of hetarylaldehydes and represented a way towards various carboxylic acid derivatives such as esters or amides. The reduction or condensation reactions could be proposed as a way for subsequent functionalization to generate the formyl electrophilic group. Thus, the electronic properties of the formed aldehyde will be vastly dependent on the position where the group is added. Portilla and co-workers designed a synthetic approach in which the **PP** ring was functionalized with a formyl group at the highly nucleophilic position 3 in a one-pot manner. By using Vilsmeier-Haack conditions, the 7-arylpyrazolo[1,5-*a*]pyrimidines **20** were successfully functionalized with a formyl group forming the expected aldehydes **135a–k** [19,21]. The reaction shows a broad substrate scope although with a slight dependence on the electronic properties of the 7-aryl group, that is, when π-excedent rings such as thiophene are in that position, the C3 carbon pyrazolic on **20** is even more nucleophilic, achieving higher yields (**135g**) compared with other π-deficient rings (**135f**) (Scheme 39).

Other formylation protocols have been produced to add the formyl group; for instance, Lindsley et al. employed a Suzuki reaction to add a vinyl group at position 3 of **136**, the reaction proceeds with a commonly used catalyst for this reaction generates **137** in 52% yield [35]. Then, the authors employed Osmium tetroxide in a catalytic amount with a radical oxidant in stoichiometric quantity to produce a diol over the vinyl group at **136**. Lastly, an excess of a strong but selective oxidant (sodium periodate) was added to obtain **138** by a oxidative diol rupture (Scheme 40a) [35]. Similarly, Li and co-workers reported a protocol to obtain formyl group at position 6 of **139** employing an ester moiety. The authors performed a reduction of **139** with a selective hydride donor (DIBAL-H) [55]. However, due to excess reductive reagent, the alcohol **141** was mainly obtained, which after purification suffered an oxidation reaction with PCC, obtaining the desired aldehyde **140** in a good yield (Scheme 40b) [55]. The authors report the reaction performance at room temperature for both protocols; the ester moiety could be less electrophilic than expected because of its electronic participation in the heteroaryl.

##### Nitration and Halogenation Reactions

Halogen atoms or nitro groups are usually added over the pyrazolic moiety of the **PP** core via electrophilic aromatic substitution reactions. Portilla and co-workers designed an MW-assisted protocol where, by using the suitable electrophile, they could functionalize position 3 of the 7-arylpyrazolo[1,5-*a*]pyrimidine **20** [38]. The conditions employed to achieve halide derivatives **142a**–**i** in high yields involve *N*-halosuccinimides (NXS) as a halogen source at room temperature for 20 min. On the other hand, nitration reactions readily occur at position 3 where, despite the conditions, nitration of the 7-aryl moiety in the substrate was not observed, achieving good yields in a short time for 3-nitroderivatives **143a**–**d** (Scheme 41).

Similarly, Gazizov and co-workers employed classic conditions for the formation of the nitronium ion, which delivered products **145a–b** in high yields from **144** [83]. The conditions show interesting tolerance despite the possible amine oxidation reactions or hydrolysis of the nitrile group in **144b** or ester on **144a** (Scheme 42a). Moreover, the authors report the employment of other sources of nitrate ions like potassium nitrate (not shown)**.** Concerning the halogenation reactions, Yamaguhi-Sasaki et al. described a reaction that differs from the common Vilsmeier-Haack conditions, developing a methodology that enables the hydroxyl/enone transformation chloride from DMPA (Scheme 42b) [78]. Commonly, this reaction allows for subsequent NAS reactions (as observed in Scheme 4).

##### Reduction Reactions

Portilla and co-workers designed a strategy that enables the formation of amines at position 6 by using 5-amino-6-(phenydiazenyl)pyrazolo[1,5-*a*]pyrimidines **44a–c**. This strategy delivers the 1,2-diamine system in **148a–c**, which could be used to synthesize other fused rings (Scheme 43a) [53]. Notably, the free amine at position 6 is not easily obtained with a common aromatic substitution. Afterward, the same research group performed the synthesis of the 3-aminoderivatives **149** through catalytic reduction of the appropriate 3-nitroderivative **143** (Scheme 43b), which was described in the previous section (see Scheme 41) [38]. Related to the previous reduction reactions, Wang and co-workers reported the obtention of the tetrahydroderivative **150** from the amide **23a** employing strong reducing conditions (Scheme 43c) [39].

## 3. Antitumor Activity

Pyrazole derivatives are involved in many medical applications and are known to be a biologically relevant scaffold [1,2,3,4,5,6,84,85]. Besides, pyrazole[1,5-*a*]pyrimidines are fused pyrazoles that have attracted particular attention in the cancer treatment field [86,87,88]. Herein, we analyze the recent publications about new molecules and their respective roles in vitro and in vivo applications, allowing us to identify the principal structure motifs for future novel uses. This section delves into the principal and recent advances of pyrazolo[1,5-*a*]pyrimidines as an antitumor scaffold in bioactive compounds, mainly by inhibiting the reproduction of cancer cells [8,57,89,90,91,92] or enzymes directly related to abnormal cell reproduction [82,93,94,95].

### Antiproliferative Activity

McCoull and co-workers [89] built up a novel procedure for macrocyclic motifs development bearing a PP core by obtaining a series of **BCL6** binders from both fragment and virtual screening. Henceforth, dislodging crystallographic water, framing new ligand protein connections, and performing a macrocyclization are actions performed to support the bioactive adaptation of the ligands. The structure-activity relationship (SAR) for **PP** (Scheme 44) indicated that the lactam carbonyl formed a noticeable hydrogen bond. Likewise, the modification in C-3 significantly changes the interaction within the enzyme, where the highest affinity was found with the nitrile group. This approach could indicate a polar interaction with an asparagine residue, which stabilizes the compound inside the enzyme. Despite the good results in SAR terms and its biological action against **BCL6**, its low selectivity decreases the potential activity of these compounds.

In 2020 Lamie et al. [91] published a novel family of pyrazolo[1,5-*a*]pyrimidines that had a great activity against the human breast adenocarcinoma cell line (**MCF-7**) and colon cancer cell line (**HTC-116**). They use conventional heating and a long reaction time (6 h) to obtain the products **158** from aminopyrazoles **156** and β-ketoesters **48** (Scheme 45a). Additionally, the authors heated the compounds **158a–c** under reflux with acetylacetone (**157**) in acetic acid, obtaining the respective 5,7-dimethylpyrazolo[1,5-*a*]pyrimidine-3-carboxamides **159a–c** (Scheme 45b). The authors [91] established that the main interactions between the *N*-heterocycle and the human **PIM-1** enzyme were hydrogen bonds due to high electronegative atoms, like N and O. Furthermore, the **PP** core planar structure and the arylamide group promote π–π interactions with active site residues. They concluded from the experimental results that **158c**, **158g**, **158h**, **159a**, and **159c** showed PIM-1 inhibitory activity in sub-micromolar concentration (Scheme 45). These compounds displayed activity with IC_50_ (µM) of 1.26, 0.95, 0.60, 1.82, and 0.67, respectively; thus, pointing out the relationship between the **PIM-1** hindrance and anticancer action against colon and breast cancer cell lines.

Chen et al. [41] published a research article where 24 pyrazolo[1,5-*a*]pyrimidines were synthesized by using the β-enaminone **19h** and 3-bromo-1*H*-pyrazol-5-amine (**7b**) as reagents. In this work, only compounds **162** and **163** exhibited a crucial activity against **B16-F10**, **HeLa**, **A549**, and **HCT-116** cancer cell lines when compare against Colchicine (Scheme 46). Two conclusions could been derived from SAR analysis; first, the presence of electro-donating groups (EDGs) as the 4-tolyl group in compound **163** results in higher biological activity in contrast EWGs. Secondly, the substitution of another EDG as the 5-indolyl group (compound **162**) increases the activity against cancer cells. Additionally, the authors proposed that compound **163** was exceptionally viable in restraining melanoma tumor development in vivo with no conspicuous poisonousness.

In 2016, Zhang and co-workers published an article in which they carried out a coupling between *N*-mustard residue with **PP** derivatives to afford 43 new compounds [8]. The compound **168** (Scheme 47) possessed antiproliferative activity with IC_50_ (µM) values of 6.023, 0.217, 6.318, 8.317, and 6.82 against the **A549**, **SH-SY5Y**, **HepG-2**, **MCF-7**, and **DU145** cell lines, respectively. For this reason, they chose this compound for further in vitro and in vivo studies. Thus, the authors conclude that **168** is a promising anticancer agent since it showed higher activity and less cytotoxicity than control drugs, Sorafenib and Cyclophosphamide. From the library created by the researchers, they established that for an exhibit potent cytotoxicity in vitro, the *N*-mustard pharmacophore and aniline moieties must be linked at 5 and 7 positions of the **PP** core, respectively (Figure 3).

In 2020, Abouzidb et al. [40] published the synthesis of some pyrazolo[1,5-*a*]pyrimidines as novel larotrectinib analogs using a reaction between β-enaminones **19** and the 4,5-disubstituted 3-aminopyrazole **169** (Scheme 48). The antiproliferative activity of compounds **170e**, **170j**, and **170k** stand out as the most active against three cancer cell lines: hepatocellular carcinoma **Huh-7**, cervical adenocarcinoma **HeLa** and breast adenocarcinoma **MCF-7**. The improvement in the biological activity came from the substitution on the 7-aryl group with methoxy function in **170j** and the higher naphthalene-ring lipophilic character in **170k**. Notably, the activity of compounds **170a–m** is independent of the substituent electronic effect on the 7-aryl group.

In 2016, Narsaiah et al. [92] published a series of pyridine-fused pyrazolo[1,5-*a*]pyrimidines **174** (Scheme 49). The pyrido[2′,3′:3,4]pyrazolo[1,5-*a*]pyrimidines **174** were prepared from the key intermediate pyrazolo[3,4-*b*]pyridine **171** and different 1,3-biselectrophilic reagents (**32**, **172** and **173**). The compounds were screened for relative global growth inhibition against five human cancer cell lines (**PC3**, **MDA-MB-231**, **HepG2**, **HeLa**, and **HUVEC**), 5-fluorouracil (5-FU) as a positive control, and DMSO as a negative control. Although the authors mention that all compounds have potential anticancer activity, only compounds **174a** and **174f** have IC_50_ values lower than the control.

Through the reaction of β-enaminones **19** with the pyrazolic ester **4**, Kumar et al. [42] described the synthesis of **175** as a building block for obtaining different amides **176a–u** having the 7-arylpyrazolo[1,5-*a*]pyrimidine fragment, of which some showed great biological activity, inhibiting the reproduction of human cervical cancer cell line (Scheme 50).

The IC_50_ results showed that compounds with better antiproliferative activity contain at least three halogen atoms. Importantly, there may be a synergy effect between aryl groups of the two amide fragments since when at least one non-halogen substituent is introduced in one of these two rings, the activity decreased substantially. This report is in concordance with the postulated by Narsaiah et al. [92] when introduce the trifluoromethyl substituent to increase the lipophilic activity of pyrazolo[1,5-*a*]pyrimidines and with this have a better global growth inhibition of cancer cell lines (see Scheme 49).

Ismailb and coworkers [57] published in 2019 an investigation focused on the inhibition of CDK2 enzyme; with this purpose, they synthesize ethyl 2-(phenylamino)-4,5-dihydropyrazolo[1,5-*a*]pyrimidine-6-carboxylate derivatives (Scheme 51). Even though the 4,5-dihydropyrazolo[1,5-*a*]pyrimidines derivatives not generated the higher inhibition of the CDK2 enzyme, the compound **178d**, and **178f** revealed the most increased activity against the four tumor cell lines (**HepG2**, **MCF-7**, **A549**, and **Caco2**). This result allows establish the crucial role of fluor atom and nitrile group in the 5-aryl-7-methyl-4,5-dihydropyrazolo[1,5-*a*]pyrimidines **178** for increased the antitumor activity of compounds.

Based on previous reports [95,96] Husseiny proposed and carried out the synthesis of the 2-(benzothiazol-2-yl)pyrazolo[1,5-*a*]pyrimidine **181** [97] (Scheme 52). The author carried out the condensation reaction between 4-(benzo[*d*]thiazol-2-yl)–*N*^3^-phenyl-1*H*-pyrazole-3,5-diamine (**179**) and diethyl ethoxymethylenemalonate (**180**) under reflux in acetic acid. Compound **181** exhibited re-markable growth inhibition activity, especially against leukemia **CCRF-CEM** and lung cancer **HOP-92** cell lines. The presence of the ethyl carboxylate group at position 6 gave rise to a notable increase in cytotoxic activity against most cancer cell lines. In this work, Husseiny links together successfully two novel antitumoral scaffolds to increase the activity of the final compound against cancer cells; besides, he shows an easy way to obtain pyrazolo[1,5-*a*]pyrimidines linked to another aromatic heterocycle with huge pharmacological importance.

## 4. Enzymatic Inhibition

Mikami et al. [82] published in 2017 the synthesis of the pyrazolo[1,5-*a*]pyrimidine **187a** (Scheme 53), orally bioavailable and selective molecule that possesses CNS drug-like characteristics—including minimal molecular weight, low topological polar surface area, and a limited number of hydrogen bond donors– which translated into excellent brain penetration with no indication of P-glycoprotein (P-gp)-mediated efflux liability.

The high efficiency of fused pyrazoles **187a** must be four different effects (Figure 4):1.Pazolo[1,5-*a*]pyrimidine core that makes π − π and CH − π interactions with the residues in the enzyme.2.The central amide linker generated the hydrogen bond with the more polar residues stabilizing the PDE2A enzyme, while the amide NH plays a crucial role in constraining the binding conformation via intramolecular hydrogen bonding to the pyrimidine nitrogen atom.3.α-Branched benzylamine portion and the *p*-CF_3_O group fits into the sits in the hydrophobic cavity in an energetically favorable orthogonal orientation.4.The substitution at position 6 combines two effects, the Van der Walls interactions with the residues and the smallest possible size.

Finally, the author establishes that a single enantiomer **187a** was more potent than a racemic compound, while **187b** was inactive.

Dowling et al. [94] published in 2017 the synthesis of the **PP** salt **190a** (and other derivatives, Scheme 54a), which exhibits improved cellular activity (Wnt DLD-1 Luciferase, IC_50_ = 50 nM), high solubility, and reduced intrinsic clearance in rat hepatocytes and human microsomes relative to the analogs that have modifications at position 4 of the aniline moiety. Supported by the X-ray crystallographic structures of CK2α with **190a**, the authors identify the importance of the primary amine since directly coordinate and order water molecules and the side-chain carbonyl group the enzyme active site. Moreover, the salt **190a** has physicochemical properties that are ideal for intravenous solution formulation, has shown strong pharmacokinetics in preclinical organisms, and exhibits a high degree of monotherapy activity in xenografts **HCT-116** and **SW-620** (Scheme 54b).

In 2020 Mathinson and co-workers [77] published the synthesis of a potent RET kinase inhibitor with >500-fold selectivity against KDR (Kinase insert Domain Receptor) in cellular assays, compound **193** (Scheme 55a). The authors identify three different substitutions at position 6 for the synthesized pyrazolo[1,5-*a*]pyrimidines family (i.e., phenyl, *p*-acetamidophenyl, and 2-thiazolyl), structures **193** and **193′** in Scheme 55b. The phenyl group was used due to the synthetic availability of substrate; however, the *p*-acetamidophenyl and 2-thiazolyl groups conferred enhanced potency against a high number of transmembrane receptors tyrosine kinases and cancer cell lines. The selectivity of **193** is because the amino group at position 5 (R^2^) favors an H-bonding with a water molecule inside the enzyme active site. The piperidine ring constrained geometry facilitates Van der Waals’s interactions with some protein residues. Finally, the researchers concentrated their efforts on the installation of polarity at R^3^ to reduce overall lipophilicity; with this objective, they replaced the methoxy group with a variety of amides, the additional hydrophilicity results in a reduction in hERG binding of up to 12-fold without a substantial loss of potency in both KIF5B-RET transfected Ba/F3 cells and the LC-2/ad cell line. Unfortunately, the most promising compound bearing (methylsulfonyl)piperazine moiety, displayed substantially reduced hERG binding and exhibited insufficient oral exposure and bioavailability, precluding advancement to in vivo efficacy studies (Scheme 55b).

In 2018 Hassan et al. [95] synthesized three different *N*-heterocyclic compounds that could inhibit tyrosine kinase in cancer cells. One family of these compounds was the **PP** derivatives **195a–g** (Scheme 56). The authors determined that the compound **195f** had the most potent inhibitory activity against the epidermal growth factor receptor (EGFR) kinase enzyme. According to the docking simulation into EGFR active site is possible to say that the high activity of **195f** is due to: first, the linked by the NH_2_ backbone to the enzyme residues by a water molecule in the pocket and second, the naphthalene ring, possibly due to π–π interactions, manages to penetrate well into the pocket and out of the cleft.

Metwally et al. [28] published the cyclocondensation reaction of 5-amino-3-cyanomethyl-1*H*-pyrazole-4-carbonitrile (**54**) with acetoacetanilide (**196**) in *N*,*N*-dimethylformamide (DMF), and few drops of acetic acid to obtain the key intermediated **197**, which is used to create new pyrazolo[1,5-*a*]pyrimidines **198** (Scheme 57). Some synthesized compounds were analyzed using MTT assays on two cancer cell lines for their cytotoxic activity (breast and cervical cancer cells). Compounds **197** and **198e–f** showed higher cytotoxicity by using doxorubicin as a reference drug. Likewise, these **PP** derivatives presented inhibitory activity against KDM (histone lysine demethylases). Authors found that heteroarylidene derivatives have the highest cytotoxic activity than arylidene derivatives. Despite this, substituted phenyl group with an EWG such as 4-Cl gave the lowest cytotoxic activity. Moreover, the most active KDM inhibitor **198e** showed that 4-folds of control triggered cell cycle arrest at the G2/M step and induced a total apoptotic effect by 10 folds more than control. These results are due to the π–π interactions between the aromatic ring and the residues of the enzyme active site.

## 5. Conclusions

Despite ample literature, the innovative discoveries regarding pyrazolo[1,5-*a*]pyrimidine scaffold in synthetic and biological fields continue strongly nowadays. The research plenty focused on synthetic transformations shows a strong preference for methods that reliably deliver highly functionalized molecules from strategic reagents to avoid the need for subsequent reaction steps. The most employed synthesis pathway is based on the interaction between 1,3-biselectrophilic substrates and 3-aminopyrazoles, with various examples in diverse applications. Careful substrates selection leads to a modulable substitution pattern in the core, affecting subsequent synthetic steps. As an illustration, malonic acid delivers two hydroxyl groups in products, whereas β-ketonitriles lead to monoamine derivatives. Other significant features of the 3-aminopyrazoles route are the feasibility to scale up the reaction using cheap reagents and that a modular functionalization may also be affordable from multicomponent strategies. Importantly, only one paper of an alternative route, based on [4 + 2] cycloaddition reactions, to access **PP** derivatives was found.

Regarding post-functionalization reactions, in recent years, a strong influence from methods of bond formation supported on metal catalysis has led to the formation of interesting derivatives and structures which could be difficult to achieve before. However, aromatic substitution is still the most common method to carry out functionalization of the pyrazolo[1,5-*a*]pyrimidine core. In this vein, the fused ring prefers the interaction with nucleophiles on the pyrimidine side and with electrophiles on the pyrazole side due to the π-deficient and π-excedent nature of these rings, respectively. The principal functionalization approach resembles aromatic electrophilic substitution on single pyrazole systems (e.g., halogenation, nitration, and formylation reactions). Lastly, given that amides provide a standard industrial method to join residues is quite popular employing this functional group as a linker with **PP** derivatives. Hence, different protocols to insert this functional group have been evaluated in recent years.

Remarkable, the *N*-heterocyclic core allows crucial modifications at C2, C3, C5, C6, and C7 positions during ring-construction or later functionalization steps. These transformations can substantially modify the biological properties of compounds such as antitumoral and enzymatic inhibitory activity. However, bases on the SAR, in most cases, the presence of halogen atoms generates a remarkable effect on the cytotoxicity of the molecule. Additionally, the π–π interactions between pyrazolo[1,5-*a*]pyrimidine ring and enzymatic pocket make possible a significant number of molecules with high anticancer potential. Nevertheless, the insertion of aliphatic motifs generates a greater affinity with some enzymes, like in Kinase insert Domain Receptor. We hope this review is useful in understanding recent advances in the synthesis and functionalization of this privileged scaffold in medicinal chemistry and help the researchers to generate new ideas for rationalistic and efficient designs of pyrazolo[1,5-*a*]pyrimidine-based medical.

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
