# Peer review of "Functional Pyrazolo[1,5-a]pyrimidines: Current Approaches in Synthetic Transformations and Uses As an Antitumor Scaffold"

_molecules, 2021, doi:10.3390/molecules26092708_

Round 1

Reviewer 1 Report

The review entitled Functional pyrazolo[1,5-a]pyrimidines: Current approaches in synthetic transformations and uses as an antitumor scaffold, reported interesting overview of the current advances (2015–2021) findings in pyrazolo[1,5-a]pyrimidines synthesis and of their functionalized derivates. The paper contains material, which is worthy of publication and of high potential interest to readers. This review is useful in understanding the recent advances in the synthesis and functionalization of this privileged scaffold in medicinal chemistry. It helps the researchers to create new ideas for rationalistic and efficient designs of pyrazolo[1,5-a]pyrimidine-based medical. The paper is clear and well written, therefore, I do recommend this work to be accepted in Molecules in his present form.

A minor recommendation, at the references 10 and 11, please insert the patent publication number to be found easier.

Author Response

Reviewer 1. Accepted in present form.

Thanks for your comments, they are really very encouraging for us. However, it is important that reviewer #1 knows that the manuscript was revised again and doubled checked for typos and mistakes. Additionally, a table containing a schematic summary of the synthesis and functionalization methods of pyridines was introduced, in order to facilitate the reader's search for the synthetic transformations of interest. Finally, the conclusions were supplemented.

Reviewer 2 Report

The Authors have approach to the pyrazolo[1,5-a]pyrimidine derivatives which have high impact in medicinal chemistry and attracted a great deal of attention in material science due to their key photophysical properties. In my opinion the manuscript deserves to publish in Molecules after a minor correction. Please check the English language again and the conclusions need to be completed. Please add the conclusions regarding the synthesis of these compounds.

Author Response

Reviewer 2. Accepted after a minor correction.

I want to express my sincere gratitude for your response and comments, they are really very encouraging for us. We are very grateful for the opportunity to provide a revised version, and feel that the quality of the manuscript has been improved as a result. Please find attached an amended version of the manuscript with changes highlighted in yellow.

Q1. Please check the English language again

R./The manuscript was revised again and doubled checked for typos and mistakes.

Q2.  The conclusions need to be completed. Please add the conclusions regarding the synthesis of these compounds.

R./The conclusions were completed based on what the reviewer #2 recommended to us.

Reviewer 3 Report

The article entitled "Functional pyrazolo[1,5-a]pyrimidines: Current approaches in synthetic transformations and uses as an antitumor scaffold" is a review describing not only the synthetic way to obtain Pyrazolo[1,5-a]pyrimidines, but also the reactions to create derivatives of Pyrazolo[1,5-a]pyrimidines.

This review clearly show a huge work of reading and extraction of the litterature of this subject between 2015 and 2021. For me it suits well for Molecules and with minor revision (few spelling mistakes and one advices to increase the readibility of the review.) it can be published. 

The advice : with this huge work done, the review is long and maybe the reader, which would like to read a specific part of the part (only to functionnalize by this synthetic way the PP core) can be lost. 

Add a summary and a scheme similar to this (enclosed), this will help the reader to find more easily what he want and also to improve the quality of the paper (even if the quality is good). this scheme and summary can be added just before the point 2. synthesis and functionnalization 

p 3 line 90 it is "where they were able" 

p 5 line 148 it is "1,3-biselectrophilic ynones"

p 6 scheme 9, what is THT ? (it is not THF ?) 

p 7 line 187, hydroxy not hidroxi

p 7 line 192 and scheme 11, the structure 36 g is not the good compared to the structure of scheme 11.

p 10 line 254, arylo not aroyl

p 12 line 299

"The synthesis proved to be efficient with yields ranging from 78 to 89%, interestingly the pyrrolic nitrogen atom from the pyrazole is able to coupling probably due to a Michael conjugated addition where the acetamide is the leaving group, after the condensation from the aldehyde 65 and the amino group (NH2) of 18"

With theses modifications, the paper can be published.

Author Response

Reviewer 3. Minor revision.

I want to express my sincere gratitude for your response and comments, they are really very encouraging for us. We are very grateful for the opportunity to provide a revised version, and feel that the quality of the manuscript has been significantly improved as a result. Please find attached an amended version of the manuscript with changes highlighted in yellow.

Q1.  Spelling mistakes. 

R./The manuscript was revised again and doubled checked for typos and mistakes.

Q2.  Advices to increase the readibility of the review.

R./ Thanks for your observation. Based on what the reviewer #2 recommended to us, a table containing a schematic summary of the synthesis and functionalization methods of pyridines was introduced to facilitate the reader's search for the synthetic transformations of interest. In addition, two subtitles (2.1.2. Multicomponent reactions and 2.1.3. Synthesis by pericyclic reactions) were introduced, both of which were missing.

Q3. Other corrections.

R./The other suggested corrections were located and resolved in detail.
